# Glomerular endothelial derived vesicles mediate podocyte dysfunction: A potential role for miRNA

N. Hill[1], D. L. Michell[2], M. Ramirez-Solano[3], Q. Sheng[3], C. Pusey[1], K. C. Vickers[2], K. J. Woollard[1]*

1 Department of Medicine, Centre for Inflammatory Disease, Imperial College London, London, United Kingdom, 2 Department of Medicine, Vanderbilt University Medical Center, Nashville, Tennessee, United States of America, 3 Center for Quantitative Sciences, Vanderbilt University Medical Center, Nashville, Tennessee, United States of America

* k.woollard@imperial.ac.uk

## Abstract

MicroRNAs (miRNA) are shown to be involved in the progression of several types of kidney diseases. Podocytes maintain the integrity of the glomerular basement membrane. Extracellular vesicles (EV) are important in cell-to-cell communication as they can transfer cellular content between cells, including miRNA. However, little is known about how extracellular signals from the glomerular microenvironment regulate podocyte activity. Using a non-contact transwell system, communication between glomerular endothelial cells (GEnC) and podocytes was characterised *in-vitro*. Identification of transferred EV-miRNAs from GEnC to podocytes was performed using fluorescence cell tracking and miRNA mimetics. To represent kidney disease, podocyte molecular profiling and functions were analysed after EV treatments derived from steady state or activated GEnC. Our data shows activation of GEnC alters EV-miRNA loading, but activation was not found to alter EV secretion. EV delivery of miRNA to recipient podocytes altered cellular miRNA abundance and effector functions in podocytes, including decreased secretion of VEGF and increased mitochondrial stress which lead to altered cellular metabolism and cytoskeletal rearrangement. Finally, results support our hypothesis that miRNA-200c-3p is transferred by EVs from GEnC to podocytes in response to activation, ultimately leading to podocyte dysfunction.

## Introduction

In the kidney, glomerular endothelial cells (GEnC) in direct contact with the bloodstream bear fenestrations which allow the passage of certain molecules through the glomerular filtration barrier. The capillary endothelial cell layer of the glomerulus is characterised by circular, transcellular pores, around 60–80 nm in diameter that occupy 30–50% of the endothelial surface [1, 2]. This makes them ideal opportunists to transport proteins, biochemicals and vesicles from the blood stream to the underlying glomerular basement membrane (GBM) and podocytes. The glomerular endothelial glycocalyx is another important layer to consider, which may regulate vascular permeability, controlling both size and charge exclusion properties [3].

**Data Availability Statement:** Data can be accessed on NCBI's Gene Expression Omnibus: GSE129883 (RNAseq) and GSE12988 (sRNAseq).

**Funding:** This study was funded by the Nadhmi Auchi Foundation, Imperial College London, Kidney Research UK (RP_019_20160303 and RP_002_20170914), the Foundation for the National Institutes of Health (HL128996, HL127173, and HL116263), and the Institute for Health Research Biomedical Research Centre (Imperial College Healthcare NHS Trust and Imperial College London). The funders had no role in study design, data collection and analysis, decision to publish, or preparation of the manuscript.

**Competing interests:** The authors have declared that no competing interests exist.

The cytoskeletal dynamics and structural plasticity of podocytes make its homeostasis fundamental for normal functioning of glomerular filtration and thus proper renal function [4]. Podocytes are terminally differentiated cells with a limited capacity to regenerate, thus podocyte dysfunction and loss closely correlate with the development of proteinuria and various glomerular diseases [5]. For example, podocyte injury resulting in podocyte foot process effacement and depletion is a hallmark of progressive diabetic kidney disease (DKD) [6]. As their complex morphology contributes to glomerular permeability, the manifestation of proteinuria is associated with marked morphological changes in these cells, as has been described in patients with diabetes [7].

Podocyte function is responsive to a variety of factors and stimuli, and a role for extracellular vesicles (EV) remains to be determined but has great potential. Originally, work by Pascual et al [8] suggests that glomerular derived EVs are present in urine. Conveying information via circulating EV is now deemed to be a *bona-fide* method of cellular communication [9], as many studies have demonstrated the functional impact of EV-miRNA in recipient cells [10]. Most interestingly, the miRNA content of EVs has been profiled in many cell types and diseases. For example, dendritic cell (DC) derived EV are differentially loaded with specific miRNAs depending on donor DC maturation status, and EV-transferred miR-222 and miR-221 were found to repress target mRNAs in recipient (acceptor) DCs [11]. Recently, FOXP3 Treg-cell-derived EVs were shown to contain miRNA distinct from other T cells and these Treg-cell-derived EV were found to suppress Th1 cells in a let-7d dependent manner [12].

Relevant to kidney disease, Wu et al demonstrated the transfer of EV from mouse primary kidney GEnC to conditionally immortalised mouse podocytes, and showed that these vesicles transferred TGFB1 mRNA, causing functional effects on the podocyte [13]. However, a detailed understanding of GEnC derived EV during steady state or cell stress, and the effects of these EV on podocyte phenotype and functions remain to be determined. More broadly, understanding the key interactions between cells in the glomerulus during kidney disease will likely provide an opportunity for identifying novel drug targets that can be used to treat or reverse pathogenicity. In order to test the effects of GEnC EV on podocyte function, we utilised an *in-vitro* co-culture model using EV derived from GEnC cultures in the steady state or under stress conditions using well described inflammatory mediators, including lipopolysaccharide (LPS) and puromycin aminonucleoside (PAN), or high glucose microenvironments representative of DKD [13–15].

## Methods

See extended methods in supplemental methods for renal cell line culture methods and full details on analytical methods used for podocyte assessment.

### Glomerular endothelial cell treatments and EV isolation

$2 \times 10^6$ GEnC were seeded onto attachment Factor coated T25 flasks and thermo switched for 3–6 days. Cells were washed, and media were replaced with Microvascular Cell Growth Kit media (Promocell) plus 5% exosome depleted FBS (Thermo Fisher Scientific). GEnC were left untreated (UT) or treated with 30mmol glucose (GLU), or 1μg/ml lipopolysaccharide (LPS) or 100μg/ml puromycin aminonucleoside (PAN) for 24 hours. Cells were counted by haemocytometric assays and supernatant subsequently centrifuged at 300g for 10 min. The supernatants were transferred to a fresh tube (Falcon) and centrifuged at 2,000g for 15 minutes. Again, the supernatants were transferred to ultracentrifugation tubes (Beckman Coulter), and centrifuged for 30 minutes at 10,000g. These steps were designed to remove cells, and cellular debris and large EV (eg. Microvesicles). The supernatants were collected, transferred to new sterile

ultracentrifugation tubes and PBS added. Equal volumes were then centrifuged at 100,000g for 70 minutes at 4˚C. EV were pelleted at 100,000g, washed and re-centrifuged to further purify EV pellets from potentially contaminating proteins. The supernatants were discarded and the concentrated EV pellets were resuspended for downstream analysis and/or experiments. Ultra-centrifugation was performed using a SW27 rotor (Beckman Instruments).

## Nanoparticle tracking analysis

Nanoparticle tracking analysis (NTA) was carried out using an LM10, according to manufac-turers instructions (NanoSight Ltd). EV were resuspended in 1ml PBS/0.1% BSA. After ensur-ing even distribution of the EV through trituration, 1 ml of solution was drawn into the syringe and inserted into the NanoSight. Data were analysed by NTA 3.0 software (Malvern Instru-ments), which was optimised to identify and track individual EV on a frame-by-frame basis.

## Transwell culture

Briefly, $2x10^5$ GEnC were seeded onto a transwell filter with a pore size of 0.4μm to prevent direct cell migration (Corning). Podocytes ($2x10^5$) were seeded onto a 12 well plate. Cells were allowed to adhere and the transwell inserts were placed on top of the podocytes (S1 Fig). Light microscopy was used to determine the quality of GEnC and podocyte monolayers. Cells were then co-cultured for 24 hours with addition of 5% EV-depleted FBS (Thermo Fisher Scientific) before transfer assays.

## RNA sequencing and analysis

To profile both small RNAs (e.g. miRNAs) and total RNA (e.g. mRNA), high-throughput sequencing libraries were generated using the NextFlex (BioO) and Ovation (NuGen) library generation kits, respectively. Prepared libraries were checked for quality control and small RNA libraries were sequenced on the NextSeq500 (Illumina) and total RNA libraries were sequenced on the NovaSeq6000 (Illumina) platforms. Bioinformatics were performed using in-house data analysis pipelines [16]. Differential expression analyses were completed using DEseq2 [17]. See 'extended methods' in supplemental methods. Data can be accessed on NCBI's Gene Expression Omnibus: GSE129883 (RNAseq) and GSE12988 (sRNAseq).

## Seahorse assays

To investigate changes in the metabolic profile of human podocytes in response to EV derived from GEnC, a Seahorse extracellular flux analyser with mitochondrial and glycolysis stress tests (Agilent) were used. See 'extended methods' in supplemental methods.

## Podocyte staining, VEGF measurement and transfection

Mitochondrial, ROS and actin cytoskeleton staining are described in 'extended methods' in supplemental methods. VEGF secretion was quantified with or without miRNA transfection by ELISA.

## Statistical analysis

Data are presented as mean ± SEM unless otherwise indicated. Statistical differences between groups were analysed using Student's t test or ANOVA (with Dunnett multiple comparisons test) for parametric data, and Mann-Whitney or Kruskal-Wallis (with Dunn's multiple com-parisons test) for non-parametric data. Data were tested for normality of distribution using the

Shapiro-Wilk normality test. All data were analysed using the GraphPad Prism 7.0 software package. P values of <0.05 were considered significant.

## Results

### Glomerular endothelial cells secrete extracellular vesicles

To examine the secretion of GEnC-derived EV in steady-state and in response to activation under relevant kidney inflammatory conditions, EV secreted from cultured GEnC were assessed *in-vitro* under specific conditions (Fig 1A and S1 Fig). Using nanoparticle tracking analysis (NTA), we quantified that GEnC secrete approximately $3x10^9$ EV per $1x10^6$ GEnC (Fig 1B–1D). To determine the impact of cellular activation on EV secretion, GEnC were activated with 1μg/ml LPS, 30mmol glucose (GLU) or 100μg/ml PAN for 24 hours in exosome-depleted FBS to represent microenvironments relevant to kidney disease, including DKD [18]. Endothelial activation was confirmed by measuring inflammatory IL-6 and IL-8 production in response to LPS stimulation (S2 Fig). Interestingly, the number of secreted EV was not affected by endothelial activation with each condition (Fig 1D). Importantly, all treatments used for endothelial activation failed to alter cellular viability (S3 Fig). Moreover, these treatments also failed to affect EV size, which were approximately 100nm in diameter, and thus likely small EV (e.g. exosomes derived from multivesicular bodies). Importantly, the total RNA content in response to treatments was also unchanged, but we did find a significant increase in total protein content in response to high glucose treatments (Fig 1E–1G). To further characterise and confirm EV purity, western blotting for EV markers was performed [19], as well as western blotting on donor GEnC as a control. CD63 and TSG101 expression confirmed the presence of EV, and GRP94 was not detected in the isolated EV, demonstrating a lack of cellular components and debris in the EV preparations (Fig 1H).

### Glomerular endothelial cell to podocyte intracellular communication by extracellular vesicles

To demonstrate transfer and uptake of GEnC-derived EV by recipient podocytes, GEnC were labelled with the fluorescent membrane probe, FM1-43X, which will be incorporated into EV during biogenesis [20]. GEnC were then washed to remove any unbound dye and co-cultured with podocytes in a transwell (TW) system (Fig 2A). After 24-hour co-culture (± LPS), uptake of labelled GEnC-derived EV by podocytes was visualised by confocal microscopy and quantified by flow cytometry. Podocyte uptake of labelled EV was demonstrated in both steady state (untreated) and pro-inflammatory (LPS activation) conditions (Fig 2B). Using an extracellular fluorescent quencher and flow cytometry, we were able to confirm the fluorescent labelled GEnC-derived EV uptake by podocytes (Fig 2C). Interestingly, we found no significant difference in podocyte uptake of GEnC-derived EV after LPS activation (Fig 2C). Podocyte uptake of EV from GEnC in steady state or upon LPS activation failed to alter podocyte viability or induce an overt inflammatory response, as measured by changes in inflammatory chemokine (CCL2) or cytokine (IL-8) secretion from steady state (untreated podocytes) (S4 Fig).

To assess transfer of GEnC-derived miRNAs to podocytes via EV, GEnC were transiently transfected with a *C.elegans* cel-miR-39 mimetic (Fig 2D). GEnC were then co-cultured in transwells (TW) with recipient podocytes (± LPS) for 24 hours and podocyte RNA was subsequently isolated and gene expression was analysed by qRT-PCR. Remarkably, cel-miR-39 was detected in recipient podocytes, which was found to be significantly increased after LPS activation of donor GEnC (Fig 2E).

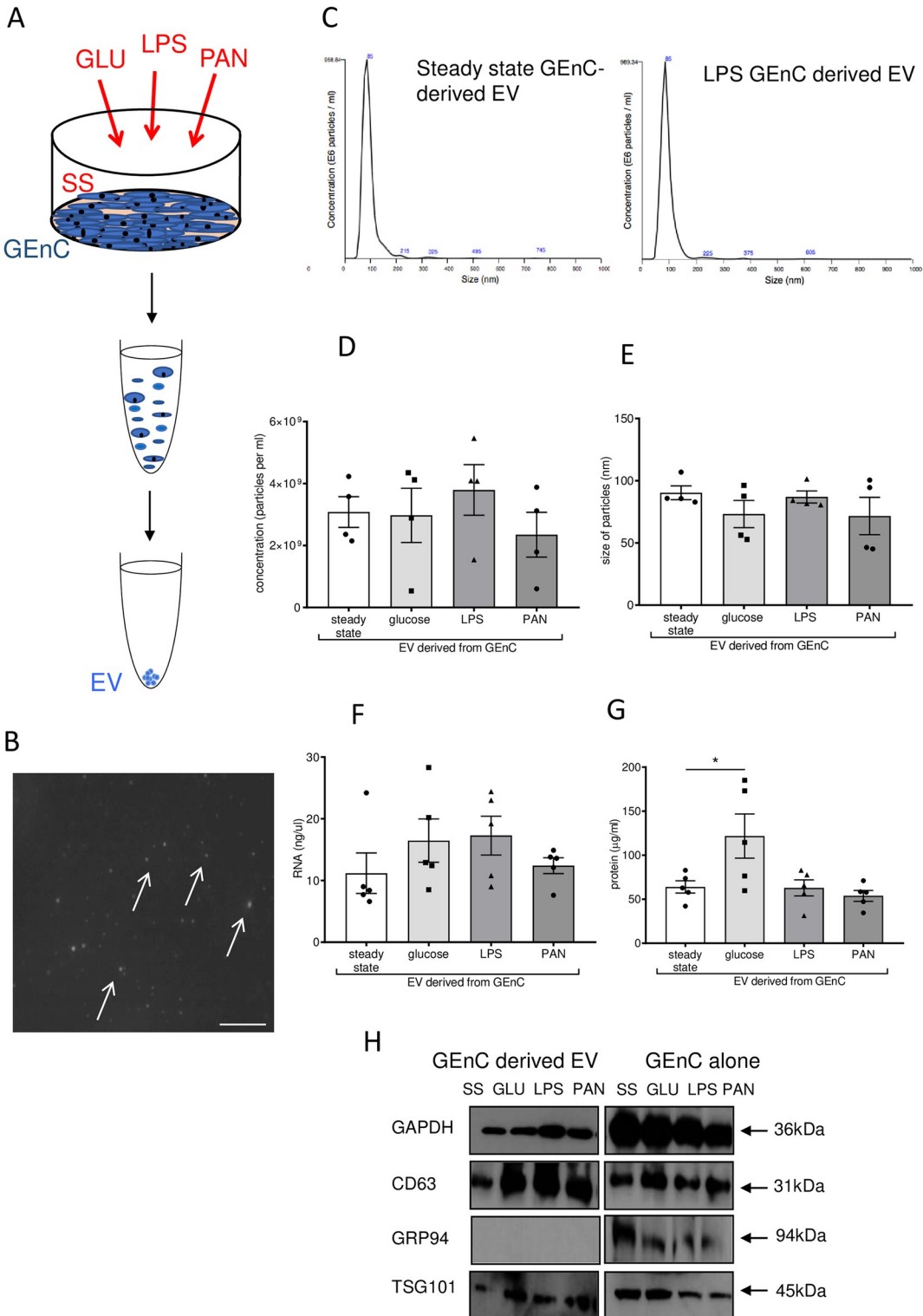

**Fig 1. Characterisation of GEnC derived EV. A)** Illustration of the UC technique for EV isolation. Briefly, GEnC cultures were supplemented with exosome free FBS and media collected 24 hour post stimulation with glucose, LPS or PAN, or GEnC were kept at steady state (control). Successive centrifugation steps at 300g, 2,000g and 10,000g removed cells, cell debris and microvesicles. UC was performed twice at 100,000g to precipitate a pure EV fraction. **B)** Snapshot of steady state GEnC derived EV as visualised by Nanosight 405nm excitation. Arrows indicated vesicles. Scale bar represents 1μm. **C)** Representative size distribution and concentration of EV isolated from steady state and LPS treated GEnC via UC, as

performed on the Nanosight LM10 instrument. **D**). The concentration and **E)** size of vesicles of from steady state, glucose, LPS and PAN GEnC derived EV n = 5. **F)** RNA and **G)** protein concentration of steady state, glucose, LPS and PAN derived GEnC EV n = 5. * represents P<0.05. **H)** Protein western blots of exosomes derived from SS (steady state), GLU (glucose), LPS or PAN treated GEnC using antibodies for GAPDH, CD63, GRP94, and TSG101 performed under reducing conditions.

## Extracellular vesicles derived from activated GEnC alter mRNA and miRNA expression in podocytes

We next investigated the impact of EV communication between GEnC and podocytes on podocyte gene expression after treatment with GEnC-derived EV from steady state or after endothelial activation. Briefly, GEnC supernatants were collected and EV were isolated using ultracentrifugation. Podocytes were subsequently treated for 24 hours with resuspended (purified) GEnC-derived EV in exosome-depleted media. Total RNA was isolated from recipient podocytes for down-stream analyses by sequencing. Small RNA (sRNA) libraries were generated for miRNA quantification by sRNA-sequencing (sRNA-seq) and total RNA libraries were prepared for mRNA analysis by rRNA-depleted total RNA-sequencing (RNA-seq). To gain further insight into the consequences of GEnC inflammatory activation on GEnC-derived EV-mediated changes to recipient podocyte gene expression, GEnC activation by LPS was completed (miRNA only).

Interestingly, EV derived from steady state (SS) GEnC failed to alter podocyte gene (mRNA) expression compared to untreated (UT) podocytes (Fig 3A). Nonetheless, significant gene (mRNA) changes were observed in podocytes treated with EV from glucose (57 in total; 57 up; 0 down) and PAN (1231 in total; 532 up; and 699 down) treated GEnC compared to podocytes treated with steady state GEnC-derived EV (Fig 3A). Most interestingly, in podocytes treated with EV from both glucose and PAN-treated GEnC, we identified many significant differentially expressed genes in podocytes that were unique for each treatment condition in GEnC. However, we also found considerable overlap between the gene sets as 48 genes were found to be upregualated in both treatments (Fig 3B). Heatmaps of the most altered (upregulated) genes in podocytes treated with EV from either glucose (GLU) or PAN-treated GEnCs, as compared to steady state (SS) GEnC-derived EVs, are presented in S5 Fig. Pathway and data enrichment analysis identified many of the upregulated genes to be associated with critical metabolic pathways, including oxidative phosphorylation and cellular respiration (Fig 3C).

Given that podocyte mRNA expression had been significantly altered in response to incubation with EVs isolated from stimulated GEnCs, we also analysed podocyte miRNA changes upon treatments. Strikingly, we identified multiple miRNAs that were significantly altered in podocytes treated with EV derived from GEnC in the steady state (SS) as compared to untreated podocytes (UT) (26 in total; 15 up; and 11 down) (Fig 3D). However, a greater number of miRNA expression changes in podocytes were found after treatment with EV from glucose (GLU) (38 in total; 27 up; 11 down), LPS (35 in total; 28 up; 7 down) or PAN-activated GEnC, compared to EV from unactivated GEnCs (Fig 3D and 3E). It should be noted that EV derived from GEnC activated by PAN caused a significant increase in many miRNAs in recipient podocytes (154 in total; 153 up; and 1 down). Interestingly, EV from Glucose, LPS and PAN-treated GEnC showed highly altered expression of treatment specific miRNAs compared to only 6 shared miRNAs between all 3 treatment groups (Fig 3E). Heatmaps of the most upregulated miRNAs from podocytes treated with GLU and PAN-stimulated GEnC-derived EV, as compared to podocytes treated with EV from resting GEnC (SS), are presented in S6 Fig. Pathway and data enrichment analysis of predicted target genes (mRNA) of the significantly altered miRNAs was completed and we identified many of the upregulated miRNAs to be associated

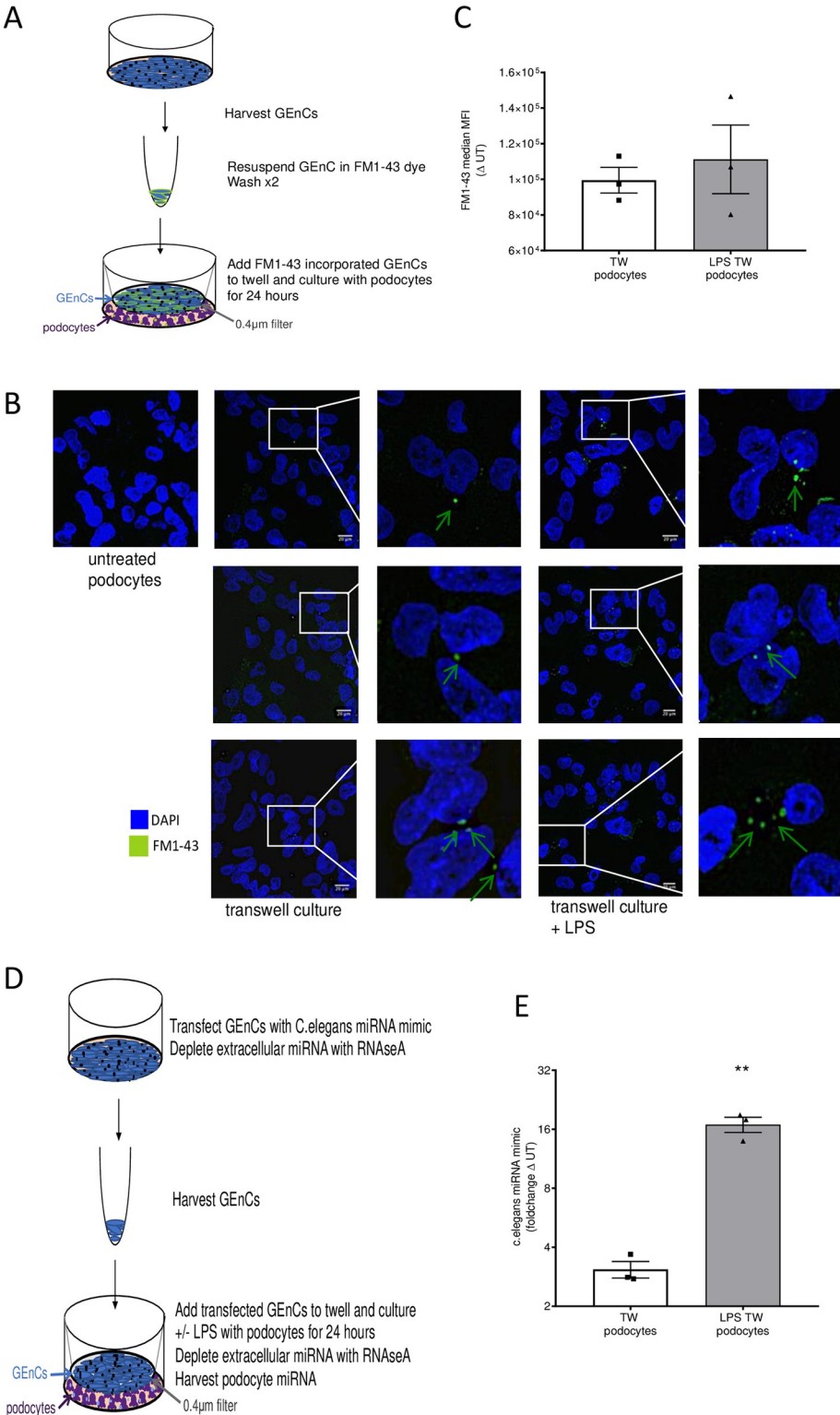

**Fig 2. Podocyte incorporation of GEnC derived EV. A)** Illustration of the transwell culture of GEnC and podocytes, allowing non-contact transfer of EV. Cells are cultured in exosome free FBS. **B)** Representative images of podocyte EV uptake post 24-hour culture with GEnC derived EV. Podocytes remained untreated or cultured with EV from GEnC exposed to FM1-43X, with or without LPS activation. GEnC were washed prior to seeding onto transwell filters, to remove any unbound dye. Confocal images were acquired on a Leica SP5 confocal microscope. Scale bar represents

10μm. Arrows indicated FM1-43X positive GEnC EV (green). Blue = podocyte nuclei **C)** Median fluorescence intensity in TW (transwell) treated podocytes with EV from steady state GEnC or LPS activated GEnC. Fluorescence was measured by flow cytometry. Backdrop suppressor was added to podocytes to remove any extracellular dye/signal. n = 3 **D)** Illustration of GEnC transfected with a *C. elegans* miRNA mimic using lipofectamine. GEnC were cultured (top) on to a 0.4μm transwell filter and cultured with podocytes (bottom) for 24 hours. RNAseA was added to the podocytes to remove any free extracellular RNA, prior to podocyte RNA isolation. **E)** Fold change (from untreated non transfected GEnC–UT) in *C. elegans* miRNA in TW (transwell) treated podocytes with EV from steady state GEnC or LPS activated GEnC. n = 4 ** represents P<0.01.

with biological processes, e.g. regulation of actin cytoskeleton, oxidative stress response, metabolic pathways, inflammatory response pathways (Fig 3F).

In summary, these results support the theory that EV from activated GEnC, not from steady state GEnC, alter recipient podocyte miRNA abundance and gene (mRNA) expression associated with metabolism.

## Extracellular vesicles derived from activated glomerular endothelial cells increase respiration and mitochondrial stress in recipient podocytes

As our molecular profiling showed alteration in podocyte metabolism after incubation with GEnC-derived EV, we next investigated the metabolic profile of podocytes in response to EV treatments using Seahorse stress tests.

GEnCs were either untreated or treated with 30 mmol glucose, 1μg/ml LPS or 100μg/ml PAN for 24 hours in EV-depleted FBS. EV were isolated and added to podocytes on a XF96 well plate in a cell ratio of 4:1. Standard concentrations of oligomycin, FCCP, and rotenone/antimycin A were injected into the cell media to analyse oxygen consumption rate (OCR) using the Seahorse extracellular flux analyser (see Methods). Analysis of cellular respiration after injection allowed quantification of basal respiration, ATP production, maximal respiration, and nonmitochondrial respiration. Proton leak and spare respiratory capacity were then calculated using these parameters.

Baseline OCR was higher in podocytes treated with EVs from glucose, LPS and PAN-treated GEnC, as compared to untreated podocytes or podocytes treated with EVs from resting (steady state) GEnC (Fig 4A). To allow visualisation of the energetic demand of podocytes in response to EV transfer, we used the mitochondrial stress test. The oxygen consumption used to meet cellular ATP demand was shown to be significantly higher in podocytes treated with EV from LPS or PAN-treatment, compared to EV from untreated GEnC (Fig 4B). To measure ATP production, the ATP synthase inhibitor oligomycin is injected into the cell media. ATP production is quantified as the difference in OCR before and after oligomycin injection. ATP production was also shown to be significantly higher in LPS or PAN GEnC EV treated podocytes compared to untreated podocytes, and significantly higher in PAN-treated GEnC-derived EV-treated podocytes compared to steady state GEnC-derived EV treated podocytes (Fig 4C). In addition to increased OCR, podocyte stress can also result in increased ECAR and glycolysis [21]. In order to investigate changes in the glycolytic profile of primary human podocytes in response to EV derived from unstimulated and stimulated GEnCs a glycolytic stress test was performed. Other than a modest increase in baseline ECAR with glucose treatment (Fig 4D), no significant difference in glycolysis was seen between podocytes treated with EV derived from steady state GEnC, as compared to podocytes treated with EVs from glucose, LPS or PAN treated GEnC (Fig 4E). In response to metabolic demands or stress signals, mitochondria are the main cellular source of energy and ROS production [22].

Mitochondrial respiration increases oxidative phosphorylation and reactive oxygen species (ROS) production, key mediators of renal damage [23, 24]. Therefore, we examined podocyte

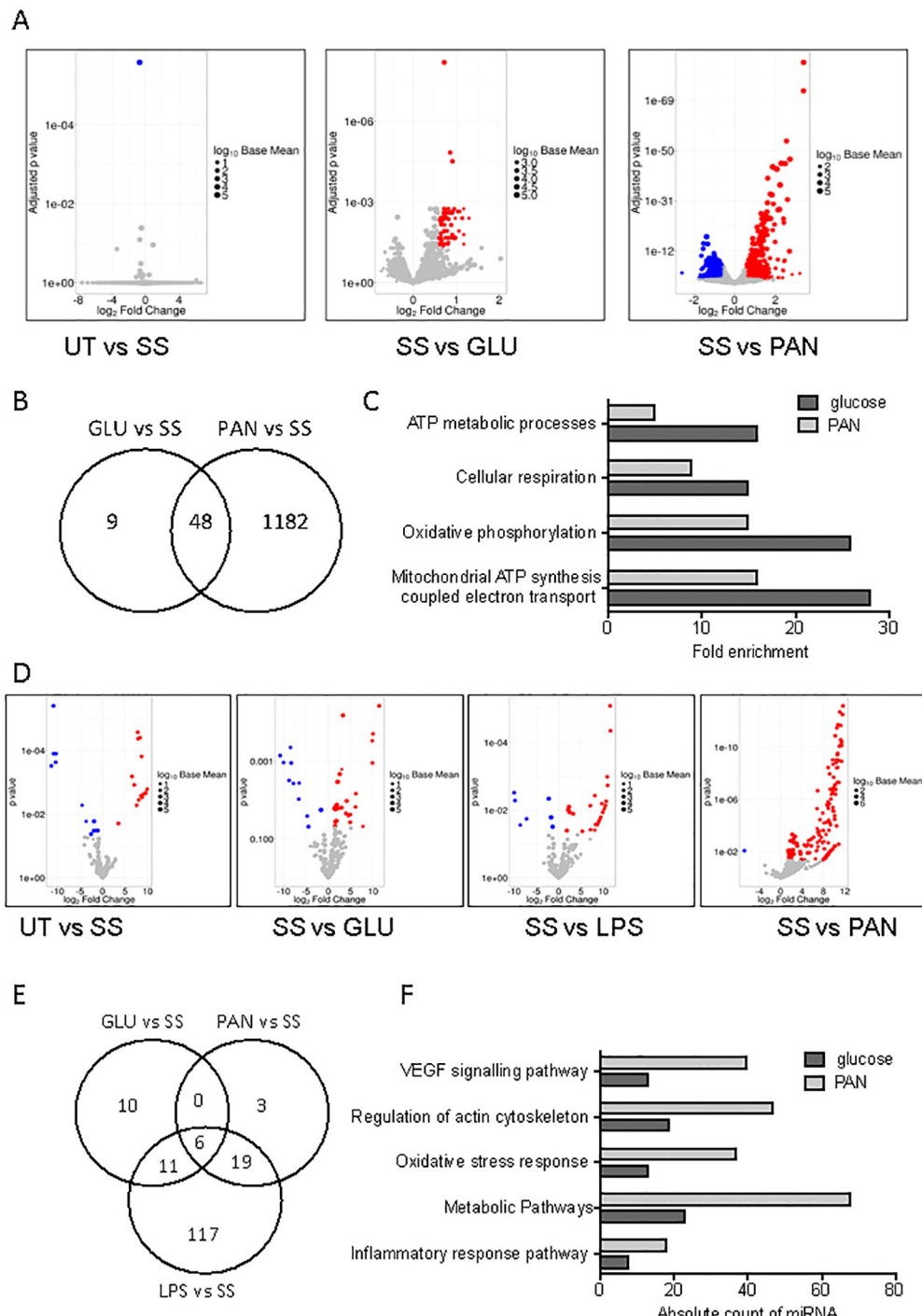

**Fig 3. GLU, LPS and PAN GEnC EV alter mRNA and miRNA expression in podocytes. A)** Volcano plots comparing podocyte mRNA fold change in UT (untreated GEnC derived EV) vs SS (steady state podocytes), SS vs GLU (glucose treated GEnC derived EV) and SS vs PAN (PAN treated GEnC derived EV). Change in miRNA expression represented as blue and red spots for significantly upregulated and downregulated miRNA respectively. Significance calculated as fold change >1.5 and P<0.05. The size of dots represents the log10 base mean value. n = 3. **B)** Venn diagram of overlapping or treatment specific significant (fold change >1.5 and P<0.05) mRNA changes in each GEnC treatment compared to steady state (SS) GEnC. **C)** GO analysis of pathways enriched from mRNA sequencing analysis (see Methods). Bar plot illustrating GO associations upregulated from SS vs GLU and SS vs PAN treatments. **D)** Volcano plots comparing podocyte miRNA fold change in UT (untreated GEnC derived EV) vs SS (steady state podocytes), SS vs GLU (glucose treated GEnC derived EV), SS vs PAN (PAN treated GEnC derived EV)

and SS vs LPS (LPS treated GEnC derived EV). Change in miRNA expression represented as blue and red spots for significantly upregulated and downregulated miRNA respectively. Significance calculated as fold change >1.5 and P<0.05. The size of dots represents the log10 base mean value. n = 3. **E)** Venn diagram of overlapping or treatment specific significant (fold change >1.5 and P<0.05) mRNA changes in each GEnC treatment compared to steady state (SS) GEnC. **F)** GO analysis of pathways enriched from mRNA sequencing analysis (see Methods). Bar plot illustrating GO associations upregulated from SS vs GLU and SS vs PAN treatments.

mitochondrial activation and ROS production in response to EV. Podocytes were seeded onto coverslips and allowed to adhere. Podocytes were incubated with EV isolated from steady state GEnC or EV from glucose, PAN or LPS-activated GEnC for 24 h. Podocytes were then washed and fixed and stained using MitoTracker. Representative images are shown (Fig 4F). The fluorescence signal (activation state) of podocyte mitochondria was significantly higher in podocytes treated with glucose-treated and PAN-treated GEnC-derived EV, as compared to steady state GEnC-derived EV (Fig 4G). We investigated mitochondrial ROS secretion using MitoSOX in podocytes treated with EVs-derived from untreated, glucose, LPS and PAN-treated GEnCs. Levels of ROS production were measured by flow cytometry, and ROS production was found to be significantly increased in podocytes treated with PAN-treated GEnC-derived EV (Fig 4G).

In summary, EV from activated GEnC increased podocyte mitochondrial respiration and this response was associated with increased mitochondrial stress and subsequently increased podocyte ROS production.

## EVs-derived from activated GEnCs mediate podocyte dysfunction—A role for miRNA-200c-3p

VEGF expression is necessary for maintaining podocyte homeostasis and survival and a reduction in VEGF expression and secretion is detrimental to kidney function [25, 26]. We therefore investigated the effects of GEnC-derived EVs on podocyte VEGF production. We found a significant decrease in VEGF secretion in podocytes treated with EVs from glucose, LPS or PAN treated GEnCs compared to podocytes treated with steady state GEnC-derived EVs (Fig 5A).

Next we investigated changes in podocyte cytoskeleton, integral for maintaining normal podocyte function [27]. Phalloidin staining was used to evaluate podocyte actin cytoskeleton. Briefly, podocytes were seeded onto coverslips and allowed to adhere for 16 hours. Podocytes were then incubated with EV isolated from steady state GEnC, or treated with EV from GEnC activated with glucose, LPS or PAN for 24 hours. Representative images are shown of podocyte cell morphology (Fig 5B), demonstrating that untreated podocytes and podocytes treated with EVs from steady state GEnC clearly express actin stress fibres (arrows). However, podocytes treated with EV derived from glucose, LPS and PAN treated GEnC showed a lack of stress fibres, thus suggesting injury associated cytoskeletal reorganisation (Fig 5B) [27].

Overall, results suggest that EVs from activated GEnCs decrease podocyte VEGF expression and alter cytoskeletal arrangement, both indicative of podocyte dysfunction associated with kidney disease.

To look specifically at the mechanism of GEnC-derived EV miRNA transfer to podocyte that likely confers decreased VEGF expression in podocytes, we experimentally tested a candidate miRNA, miR-200c-3p, that was identified in our molecular screen and examined its effects on podocyte VEGF expression. VEGF is predicted to harbour a conserved miR-200c-3p target site within its 3' untranslated region (www.targetscan.org) and previous literature has reported that miR-200C-3p overexpression decreases VEGF secretion in HELA cells [28]. Moreover, our miRNA sequencing data suggests EV from PAN-treated GEnC increased podocyte expression of miR-200c-3p (Fig 5C and S6 Fig). To determine if miR-200c-3p alters VEGF expression in podocytes, we transfected podocytes with miR-200c-3p mimetic. Overexpression of miR-200C-3p in

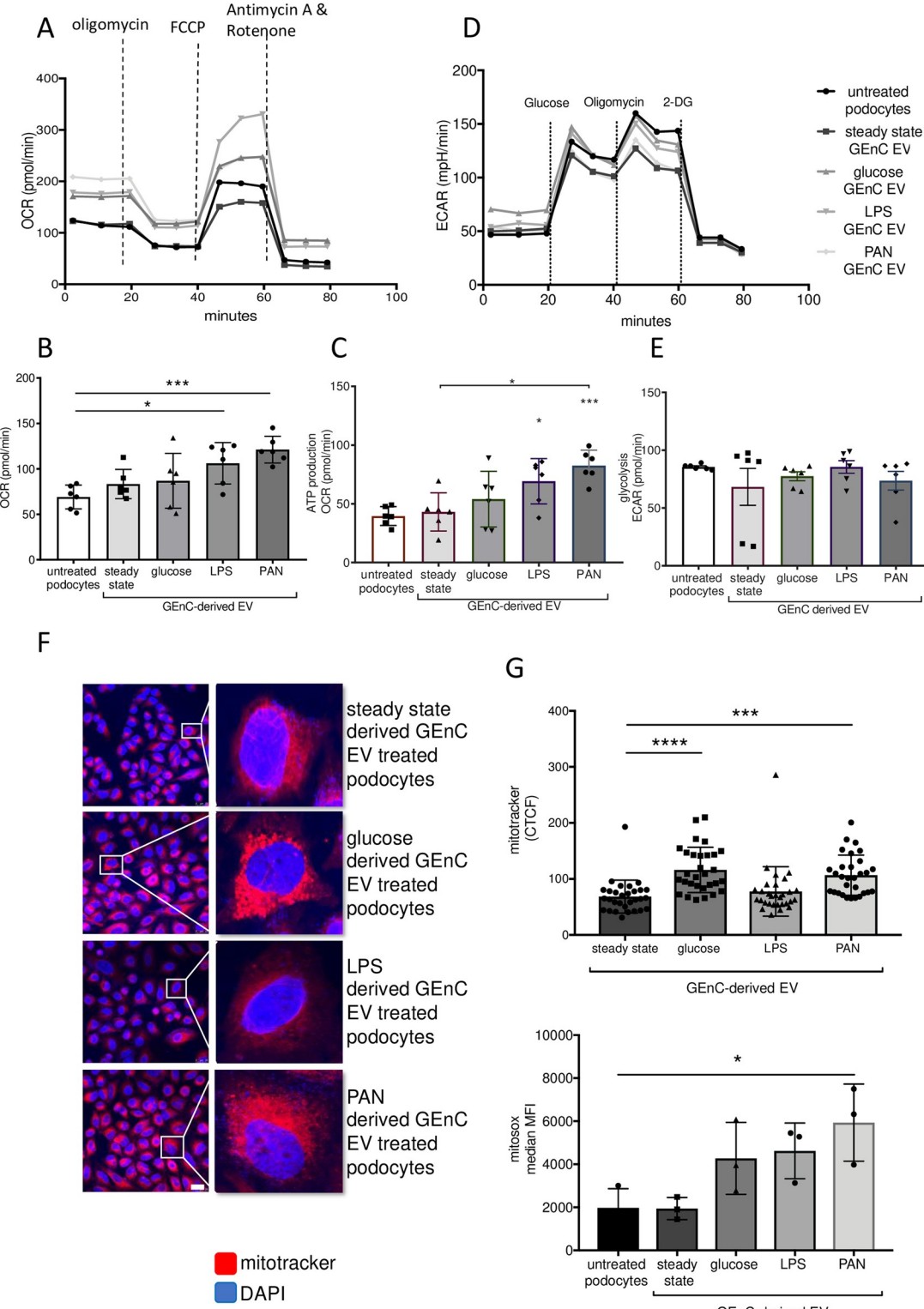

**Fig 4. Measurement of OCR of podocytes stimulated with steady state, glucose, LPS or PAN derived GEnC EV. A)**
Podocytes remained untreated or were treated with steady state GEnC EV, glucose treated GEnC EV, LPS treated GEnC E or
PAN treated GEnC EV for 24 hours. OCR (oxygen consumption rate) was measured in x96 seahorse extracellular flux analyser.
Oligomycin (1µm), FCCP (2µM), and antimycin A/rotenone (0.5µM) were injected at indicated time points. n = 6. **B)** Baseline
OCR for untreated podocytes or podocytes treated with steady state GEnC EV, glucose treated GEnC EV, LPS treated GEnC EV

or PAN treated GEnC EV for 24 hours. n = 6. **C)** Baseline ATP production for untreated podocytes or podocytes treated with steady state GEnC EV, glucose treated GEnC EV, LPS treated GEnC EV or PAN treated GEnC EV for 24 hours. n = 6. **D)** ECAR profile over time with same treatments as in A) with injection of 1μM oligomycin and 50mM 2-deoxyglucose (2- DG) as indicated. n = 6. **E)** Glycoloysis measurements at baseline after podocyte treatments with EV or at baseline. n = 6. **F)** Representative images of mitochondrial activity using 100nM MitoTracker (red) and nuclei counterstained with DAPI (blue) detected with fluorescence microscopy. Scale bar = 25 μm. **G)** Quantitative mitochondrial activation with 100nM MitoTracker after podocyte incubation with EV from steady state GEnC, glucose treated GEnC, LPS treated GEnC or PAN treated GEnC. Each dot represents 1 mitochondrion from n = 3 experiments. **H)** Flow cytometry measuring podocyte intracellular ROS levels using 5μM mitoSOX after podocyte incubation with EV from steady state GEnC, glucose treated GEnC, LPS treated GEnC or PAN treated GEnC. n = 3. *, ***, **** represents P<0.05, P<0.001 or P<0.0001 respectively, groups were compared using a one-way Anova.

podocytes caused a >50% decrease in podocyte VEGF protein levels compared to non-transfected podocytes, while the control (mimetic) treated podocytes were unchanged (Fig 5D). Interestingly, transfected podocytes with another miRNA, miR-29c-3p, which was also unregulated in response to EV with sRNA-seq (S6 Fig), failed to change podocyte VEGF secretion (S7 Fig).

In summary, these data strongly suggest that EVs containing miR-200c-3p from activated GEnC (by glucose and PAN) downregulate podocyte VEGF secretion which could contribute to renal disease.

## Discussion

These data add to the relatively limited pool of published knowledge on the response of podocytes to EV. In 2007, it was first described that EV could transfer small components to recipient cells in mast cell lines *in-vitro* [29]. The work described here enhances this evidence, suggesting, that podocytes can uptake EV from unstimulated and stimulated GEnC, resulting in changes in podocyte function. A number of key conclusions can be drawn from this work. Human podocytes uptake EV derived from GEnC. Approximately $3x10^9$ EV are secreted from $1x10^6$ conditionally immortalised human GEnC whether untreated, or treated with 30mmol glucose, 1μg/ml LPS or100μg/ml PAN. Human podocyte miRNA and mRNA gene expression were altered after 24-hour incubation with GEnC-derived EV. EV derived from stimulated GEnC cause podocyte actin dysregulation, a decrease in VEGF expression and a change in mitochondrial function.

We show multiple lines of evidence for podocyte EV uptake. The use of lipid dyes is a well-established method to detect EV transfer between cells *in-vitro* [30]. The FM1-43X lipid dye demonstrated uptake of GEnC EV by podocytes and this was proven by flow cytometry and confocal microscopy. GEnC were transfected with a *C.elegans* miR-39 miRNA mimic and further transwell cultured with podocytes for 24 hours. Podocytes incorporated the *C.elegans* miR-39 miRNA mimic from exogenous transfer. This proved to be a successful technique to track transfer of miRNA from cell to cell, but we cannot be conclusive that this was through EVs and other possible methods of miRNA transfer including protein-bound miRNAs and apoptotic bodies may be possible. Indeed, EV may also carry other components such as nucleic acids, proteins, and lipids [9, 11], which may also induce recipient cellular responses or that some components of endothelial activators (LPS, Glucose etc) may be carried over by EVs. Further work will be needed to explore these possibilities, but nevertheless our data does show that extracellular communication can occur between glomerular endothelial cells and podocytes.

Current functional studies of EV have some limitations, such as the diversity of methods used for exosome isolation. EV can be enriched from cell culture media via a variety of methods [31]. The different techniques can slightly alter the size, specificity, number of EV and EV content [32]. Due to the large number of different components that can be carried by EV and the fact that each miRNA can regulate many different signalling pathways, it is difficult to gain an understanding of the absolute function of EV on recipient cell gene expression. There has

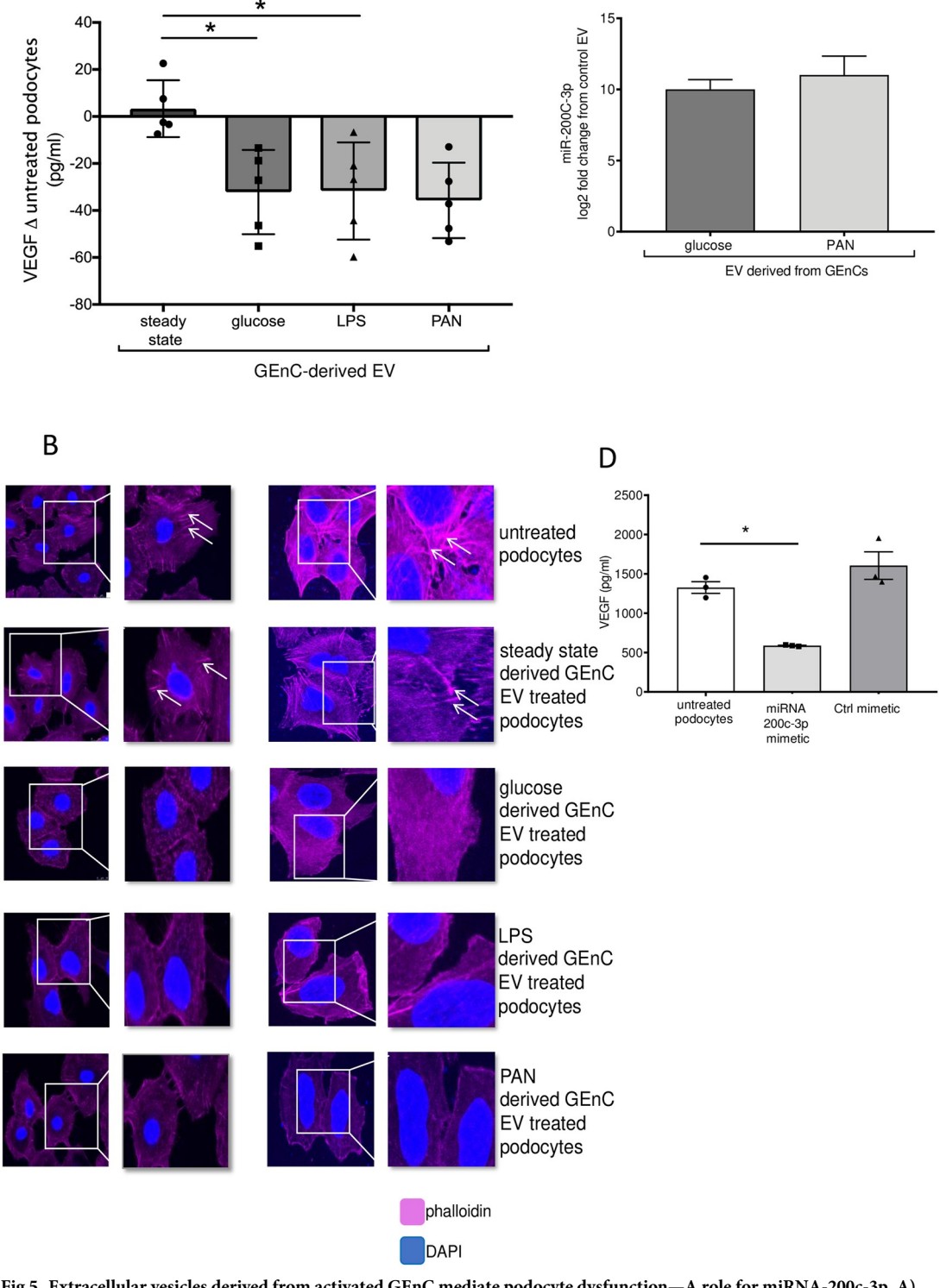

**Fig 5. Extracellular vesicles derived from activated GEnC mediate podocyte dysfunction—A role for miRNA-200c-3p. A)** Podocytes were treated for 24 hrs with EV from steady state GEnC, glucose treated GEnC, LPS treated GEnC or PAN treated GEnC and VEGF secretion measured by ELISA. Presented as change from untreated podocytes (steady state). n = 6. **B)** Representative images of actin microfilaments detected using phalloidin staining with fluorescence microscopy. Nuclei were stained with DAPI. Podocytes were treated for 24 hrs with EV from steady state GEnC, glucose treated GEnC, LPS treated GEnC

or PAN treated GEnC. Scale bar represents 25μm. Arrows indicate stress fibres during steady state. **C)** Increased expression of miRNA-200c-3p in podocytes treated with EV derived from GEnC treated with glucose or PAN versus podocytes treated with steady state GEnC derived EV (from miRNA sequencing) **D)** Effect of miR-200c-3p transfection on podocyte VEGF sprotein expression as measured by ELISA. Negative control mimic was used as a control. n = 3. *, ** and *** represents P<0.05, P<0.01, P<0.001 respectively, groups were compared using an unpaired t-test.

also been discrepancy as to whether the number of EV transferred between cells can carry enough components, (e.g. miRNA) to have an effect on the recipient cell. Chevillet et al quantified the number of exosomes from five diverse sources and from their technique suggested that, on average, there was far less than one molecule of a given miRNA per exosome [33]. However, many studies have reported functional effects of EV miRNA and mRNA *in-vitro* and *in-vivo* [34, 35].

Here a standard ultracentrifugation technique was used, identifying a pure vesicle fraction. It is possible that using different techniques could isolate slightly different pools of EV and miRNA, which is why we decided to confirm podocyte functional response to EV transfer. We show that EV from stimulated GEnC cause increased podocyte mitochondrial respiration, but only EV derived from glucose treated GEnC altered ECAR at baseline, with no changes in glycolysis. These results agree with previously published studies, suggesting that podocytes predominantly rely on mitochondrial respiration for ATP production [36] and that podocyte OCR is increased in abnormal conditions [37]. The bioenergetics of podocytes *in-vitro* were successfully measured here, although one must consider that in the kidney, podocytes are surrounded by other adjacent cells, which may additionally impact cellular respiration [38]. Therefore, taking these studies *in-vivo* would supplement the results.

As mitochondrial function was altered in EV treated podocytes, the effect of GEnC EV on podocyte mitochondrial activation and ROS secretion was examined. Both were found to be upregulated in podocytes treated with EV from stimulated GEnC, confirming that EV derived from stimulated GEnC alter podocyte mitochondrial function. Phalloidin effectively stains actin filaments for visualisation of cytoskeleton. Using this technique, the present data suggest that EV derived from stimulated GEnC alters podocyte cytoskeletal arrangement. The decrease in podocyte VEGF expression post transfection with miR-200c-3p supports the idea that EV increased miR-200c-3p expression leading to downregulation of VEGF secretion. A loss of regular actin arrangement and VEGF synthesis resulting from EV mediated podocyte injury may contribute to podocyte damage favouring the development of renal disease.

Exosomal miRNA can be used as non-invasive biomarkers to indicate disease states. They have been profiled to aid clinical diagnosis [39]. The miRNA and mRNA presented here could be used as markers for renal disease. The EV sequencing results could further add to the previous pool of data [40]. Once RNA overexpression caused by EV transfer is functionally studied, it can be applied to mouse models for further understanding of RNA transfer in the kidney.

Due to the nature of the GFB with the intense force, continual flow of solutes and water, and close proximity of GEnC to podocytes, in an *in-vivo* renal disease setting EV from disturbed GEnC could transverse through the GBM and be taken up by podocytes, resulting in altered podocyte functional responses. This premise of EV traversing through the GBM is supported by previous work demonstrating that after intravenous injection, vesicles are found in the urine, implying the in-vivo transfer of EV through the GBM [41].

In conclusion, we show that 1) GEnC EV can be taken up by podocytes, 2) EV from activated GEnC alter podocyte mRNA and miRNA, 3) EV from activated GEnC cause increased podocyte basal respiration leading to mitochrondrial stress, 4) EV from activated GEnC can cause a loss of actin stress fibers, 5) EV from activated GEnC can cause a decrease in podocyte VEGF secretion and 6) GEnC EV can cause increased expression of podocyte miR-200c-3p

resulting in a decrease in podocyte VEGF production. These findings show that activated GEnC cause functional changes to podocytes via EV uptake. Further research that explores podocyte specific responses to GEnC EV and other cell-cell cross-talk will provide mechanistic clues that underlie a variety of renal diseases including DKD and glomerulonephritis, thus providing novel avenues for therapeutic intervention.

## Supporting information

**S1 Fig. Illustration of the transwell culture system.** Illustration of the transwell culture of GEnC and podocytes, allowing both cell types to communicate through the transfer of EV. Cells are plated and cultured for 7 days. GEnC are seeded onto a 0.4μm filter. Podocytes are seeded onto the bottom of the well.
(PDF)

**S2 Fig. Impact of LPS incubation on GEnC inflammatory cytokine secretion.** GEnC were treated with 1μg/ml LPS for 6 or 24 hours. Supernatants of LPS treated GEnC were collected. IL-6 and IL-8 secretion was measured by ELISA and compared to untreated GEnC. At least 4 readings of experiments were performed in duplicate.
(PDF)

**S3 Fig. GEnC viability post 24-hour stimulation.** GEnC were left untreated or stimulated with 30 mmol glucose, 1ug/ml LPS, 100ug/ml PAN for 24hours. Cells were counted using a haemocytometer and compared to untreated cells for viability. n = 6.
(PDF)

**S4 Fig. Podocyte CCL2 and IL-8 secretion after EV exposure.** Podocyte supernatant was collected 24-hour post incubation with GEnC EV. Podocytes remained untreated or were incubated with EV from steady state, glucose, LPS or PAN treated GEnC. CCL2 and IL-8 expression was measured by ELISA. Data presented as change in expression from untreated podocytes. n = 7.
(PDF)

**S5 Fig. Heatmaps represent the top (fold change, P<0.01) 35 mRNA which have been selected for SS vs GLU and SS vs PAN comparisons for GEnC EV treated podocytes.**
(PDF)

**S6 Fig. The top (fold change, P<0.01) miRNA have been selected to create heatmaps.** SS vs GLU (top 27) and SS vs PAN (top 40) comparisons for GEnC EV treated podocytes.
(PDF)

**S7 Fig. Effect of miR-29c-3p transfection on podocyte VEGF protein expression as measured by ELISA.** Negative control mimic was used as a control. n = 8.
(PDF)

**S1 Methods.**
(PDF)

**S1 Data.**
(PPTX)

## Author Contributions

**Conceptualization:** N. Hill, C. Pusey, K. C. Vickers, K. J. Woollard.

**Data curation:** N. Hill, D. L. Michell, K. C. Vickers.

**Formal analysis:** N. Hill, D. L. Michell, M. Ramirez-Solano, Q. Sheng, K. C. Vickers.

**Funding acquisition:** C. Pusey, K. J. Woollard.

**Investigation:** K. J. Woollard.

**Project administration:** K. J. Woollard.

**Supervision:** K. J. Woollard.

**Writing – original draft:** N. Hill, K. J. Woollard.

**Writing – review & editing:** D. L. Michell, C. Pusey, K. C. Vickers, K. J. Woollard.

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
