## [Decision Letter · Decision Letter 0]

6 Dec 2019

PONE-D-19-29233

Glomerular endothelial derived vesicles mediate podocyte dysfunction via extracellular transfer of miRNA-200c-3p

PLOS ONE

Dear Dr. Woollard,

Thank you for submitting your manuscript to PLOS ONE. After careful consideration, we feel that it has merit but does not fully meet PLOS ONE’s publication criteria as it currently stands. Therefore, we invite you to submit a revised version of the manuscript that addresses the points raised during the review process.

Among the comments of the reviewers, you will answer carefully the first and the second remarks of reviewer 1 and the last remark of reviewer 2. It should also be recalled that PLOS ONE now requires that submissions reporting blots or gels include original, uncropped blot/gel image data as a supplement or in a public repository.

We would appreciate receiving your revised manuscript by Jan 20 2020 11:59PM. To enhance the reproducibility of your results, we recommend that if applicable you deposit your laboratory protocols in protocols.io, where a protocol can be assigned its own identifier (DOI) such that it can be cited independently in the future. For instructions see: http://journals.plos.org/plosone/s/submission-guidelines#loc-laboratory-protocols

We look forward to receiving your revised manuscript.

Kind regards,

Jean-Claude Dussaule

Academic Editor

PLOS ONE

Journal Requirements:

Please ensure that your manuscript meets PLOS ONE's style requirements, including those for file naming. The PLOS ONE style templates can be found at http://www.plosone.org/attachments/PLOSOne_formatting_sample_main_body.pdf and http://www.plosone.org/attachments/PLOSOne_formatting_sample_title_authors_affiliations.pdf PLOS ONE now requires that authors provide the original uncropped and unadjusted images underlying all blot or gel results reported in a submission’s figures or Supporting Information files. This policy and the journal’s other requirements for blot/gel reporting and figure preparation are described in detail at https://journals.plos.org/plosone/s/figures#loc-blot-and-gel-reporting-requirements and https://journals.plos.org/plosone/s/figures#loc-preparing-figures-from-image-files. When you submit your revised manuscript, please ensure that your figures adhere fully to these guidelines and provide the original underlying images for all blot or gel data reported in your submission. See the following link for instructions on providing the original image data: https://journals.plos.org/plosone/s/figures#loc-original-images-for-blots-and-gels.In your cover letter, please note whether your blot/gel image data are in Supporting Information or posted at a public data repository, provide the repository URL if relevant, and provide specific details as to which raw blot/gel images, if any, are not available. Email us at plosone@plos.org if you have any questions. We note that you have included the phrase “data not shown” in your manuscript. Unfortunately, this does not meet our data sharing requirements. PLOS does not permit references to inaccessible data. We require that authors provide all relevant data within the paper, Supporting Information files, or in an acceptable, public repository. Please add a citation to support this phrase or upload the data that corresponds with these findings to a stable repository (such as Figshare or Dryad) and provide and URLs, DOIs, or accession numbers that may be used to access these data. Or, if the data are not a core part of the research being presented in your study, we ask that you remove the phrase that refers to these data. Please upload a copy of Figure 4H, to which you refer in your text on page 14. If the figure is no longer to be included as part of the submission please remove all reference to it within the text.Thank you for stating the following in the Acknowledgments Section of your manuscript:'This work was funded by a Nadhmi Auchi Foundation studentship and ImperialCollege Dean’s internship award to NH. Grants from Kidney Research UK(RP_019_20160303, RP_002_20170914) and BHF (PG/18/41/33813) alsosupport the Woollard lab. Moreover, this work was supported by awards from theNational Heart, Lung and Blood Institute, National Institutes of Health (USA) toKCV (HL128996, HL127173, and HL116263). We acknowledge a contributionfrom the National Institute for Health Research Biomedical Research Centre basedat Imperial College Healthcare NHS Trust and Imperial College London.'We note that you have provided funding information that is not currently declared in your Funding Statement. However, funding information should not appear in the Acknowledgments section or other areas of your manuscript. We will only publish funding information present in the Funding Statement section of the online submission form.Please remove any funding-related text from the manuscript and let us know how you would like to update your Funding Statement. Currently, your Funding Statement reads as follows:'The author(s) received no specific funding for this work'

Reviewers' comments:

Reviewer's Responses to Questions

**Comments to the Author**

1. Is the manuscript technically sound, and do the data support the conclusions?

Reviewer #1: Yes

Reviewer #2: Partly

2. Has the statistical analysis been performed appropriately and rigorously? 

Reviewer #1: Yes

Reviewer #2: Yes

3. Have the authors made all data underlying the findings in their manuscript fully available?

Reviewer #1: Yes

Reviewer #2: Yes

4. Is the manuscript presented in an intelligible fashion and written in standard English?

Reviewer #1: Yes

Reviewer #2: Yes

5. Review Comments to the Author

Reviewer #1: Comments:

The authors inquire about the role played by extracellular vesicles (EV) trafficking between two different types of cells in the progression of kidney disease. To this aim, they use a non-contact in vitro transwell system to characterise glomerular endothelial cell to podocyte communication. Using this experimental setting, they try to get more insights into how extracellular signals from the glomerular microenvironment regulate podocyte activity.

This manuscript describes carefully performed in vitro work which demonstrates that upon injury, EVs are secreted by glomerular endothelial cells and up taken by adjacent podocytes. This mechanism leads to alteration in podocyte mRNA and miRNA content and subsequent changes to podocyte functions.

The experimental design is well set up and the conclusions are presented in an appropriate fashion and are supported by the data.

Minor issues:

1) However, there is no in vivo relevance which support these findings and no evidence this extracellular transfer is taking place in the injured glomerulus. By quoting previous work (ref 41), the authors argue their described extracellular trafficking can occur in vivo since EVs, intravenously injected in the mouse, were found in the urine, implying active in vivo transfer through the glomerular basal membrane.

In the present work, the authors demonstrated they can efficiently produce and isolate EVs from quiescent and injured glomerular endothelial cells, i.e. 3.1010 EVs per 1.106 cells. They also demonstrated uptake of EVs by podocytes using the fluorescent membrane probe FM1-43X. One can argue they could have produce high levels of fluorescently-labelled EVs, systemically injected them in the mouse and check for FM1-43X staining in podocytes.

2) In Figure 2 panels D and E, glomerular endothelial cells were transfected with C.elegans Cel-mir-39 mimetic and further transwell cultured with podocytes. Podocyte incorporated the miRNA which proved successful exogenous transfer.

However, Panel E shows that Cel-mir-39 was significantly increased in the LPS-treated condition. It is not clear how that happened since Figure 1 showed that the number of EVs was not affected by endothelial cell activation (LPS, Glucose and PAN). Although it is clear that the EV content changes from quiescent to activated endothelial cells as it is able to alter podocyte mRNA and miRNA, why would it this affect Cel-mir-39 abundance?

3) There is no error bars in Figure 5C although the analysis seems to have been performed in triplicate for each conditions (Supplemental Figure 6).

Reviewer #2: This manuscript shows in-vitro experiments describing the glomerular endothelial cells to podocyte communication via extracellular vesicles in a variety of conditions and focuses on miRNA transfer between these cells.

The manuscript is well written; the background is well described. Experimental methods are also well described. Many data are shown in order to confirm the hypothesis.

I have a few minor comments:

- On the one side, focus has been put on extracellular vesicles. In order to display all the aspects of cell-to-cell communications, the authors should also describe the other types of transfer (protein-boud miRNAs, apoptotic bodies). In this line, authors should specify if cel-miR-39 might be transferred by these mechanisms instead of EV. If their experiments cannot be conclusive they should add this limitation in their Discussion Section.

- On the other side, focus has been put on one specific component of EV, miRNAs. However, to the best of my knowledge, EV also carry other component such as nucleic acids, proteins, and lipids, all of which might induce various cellular response in the recipient cells. Authors should discuss this point. In the same line, might EV carry the treatments that GenC received (glucose, PAN, LPS)? In this case the effects seen on recipient cells could also be due to treatment transfer, which is also interesting if possible. Please discuss this point with appropriate references.

- Is the chosen pore size of the transwell system appropriate to reflect glomerular filtration barrier? Please justify this in the Methods Section.

- Results regarding the role of miRNA-200c-3p are not sufficient in my opinion in order to establish a causal relationship between EV transfer of miRNA-200c-3p and VEGF production. First, the amount of miRNA-200c-3p transferred by EV on the one side, and by transfection on the other side, is not specified. The amounts are likely to be very different. Second, other components of EV might induce the decrease in podocyte VEGF production. Finally, podocyte impairment by either mechanism might result in VEGF decrease, rather than being the consequence of VEGF decrease. For these reasons, I suggest to authors to discuss this point, and to remove “via extracellular transfer of miRNA-200c-3p” from the title of the article, which is too assertive compared to the actual results of the experiments. The title might be “glomerular endothelial derived vesicles mediate podocyte dysfunction: a potential role for miRNAs” for example.

6. PLOS authors have the option to publish the peer review history of their article (what does this mean?). If published, this will include your full peer review and any attached files.

Reviewer #1: No

Reviewer #2: Yes: Nahid Tabibzadeh

---

## [Author Response · Author response to Decision Letter 0]

17 Jan 2020

Thank you to editorial board and reviewers for their time taken to review our paper. We have now addressed all editorial concerns and journal requirements. We have added point-by-point rebuttal to reviewers comments below. 

Reviewer #1: Comments:

The authors inquire about the role played by extracellular vesicles (EV) trafficking between two different types of cells in the progression of kidney disease. To this aim, they use a non-contact in vitro transwell system to characterise glomerular endothelial cell to podocyte communication. Using this experimental setting, they try to get more insights into how extracellular signals from the glomerular microenvironment regulate podocyte activity.

This manuscript describes carefully performed in vitro work which demonstrates that upon injury, EVs are secreted by glomerular endothelial cells and up taken by adjacent podocytes. This mechanism leads to alteration in podocyte mRNA and miRNA content and subsequent changes to podocyte functions.

The experimental design is well set up and the conclusions are presented in an appropriate fashion and are supported by the data.

Thank you for the time taken to review our paper and for the encouraging comments in your review. We have added a rebuttal to each minor issue below.

Minor issues:

1) However, there is no in vivo relevance which support these findings and no evidence this extracellular transfer is taking place in the injured glomerulus. By quoting previous work (ref 41), the authors argue their described extracellular trafficking can occur in vivo since EVs, intravenously injected in the mouse, were found in the urine, implying active in vivo transfer through the glomerular basal membrane.

In the present work, the authors demonstrated they can efficiently produce and isolate EVs from quiescent and injured glomerular endothelial cells, i.e. 3.1010 EVs per 1.106 cells. They also demonstrated uptake of EVs by podocytes using the fluorescent membrane probe FM1-43X. One can argue they could have produce high levels of fluorescently-labelled EVs, systemically injected them in the mouse and check for FM1-43X staining in podocytes.

Thank you for your comments. We agree that we do not show direct in-vivo relevance. Our work establishes an in-vitro based method to show endothelial and podocyte transfer of endothelial vesicles. We are now applying for grants to complete work to show direct in-vivo efficacy, but this is out of scope for the current manuscript. 

2) In Figure 2 panels D and E, glomerular endothelial cells were transfected with C.elegans Cel-mir-39 mimetic and further transwell cultured with podocytes. Podocyte incorporated the miRNA which proved successful exogenous transfer.

However, Panel E shows that Cel-mir-39 was significantly increased in the LPS-treated condition. It is not clear how that happened since Figure 1 showed that the number of EVs was not affected by endothelial cell activation (LPS, Glucose and PAN). Although it is clear that the EV content changes from quiescent to activated endothelial cells as it is able to alter podocyte mRNA and miRNA, why would it this affect Cel-mir-39 abundance?

Thank you for this insightful comment. We were also intrigued by the observation that LPS treated endothelial cells transferred more Cel-mir-39 even though, as rightly pointed out, LPS did not seem to increase total number of glomerular endothelial secreted EVs or transfer (as shown in by FM1-43, Fig2C). We do not have a clear answer as to why LPS would affect Cel-mir-39 abundance. Our current working hypothesis is that LPS may increases some miRNA EV loading and/or transfer of miRNA transfer and EV from endothelial cell donors to podocyte recipients. These are difficult experiments to design with conclusive results and as such is work in progress and outside the scope of the current manuscript. 

3) There is no error bars in Figure 5C although the analysis seems to have been performed in triplicate for each conditions (Supplemental Figure 6).

Apologise, this was an oversight on our part. Thank you for bringing this to our attention. Error bars have now been added to Figure 5C. 

Reviewer #2: This manuscript shows in-vitro experiments describing the glomerular endothelial cells to podocyte communication via extracellular vesicles in a variety of conditions and focuses on miRNA transfer between these cells.

The manuscript is well written; the background is well described. Experimental methods are also well described. Many data are shown in order to confirm the hypothesis.

Thank you for the time taken to review our paper and for the encouraging comments on our manuscript, we have added a rebuttal to each minor comment below. 

I have a few minor comments:

- On the one side, focus has been put on extracellular vesicles. In order to display all the aspects of cell-to-cell communications, the authors should also describe the other types of transfer (protein-boud miRNAs, apoptotic bodies). In this line, authors should specify if cel-miR-39 might be transferred by these mechanisms instead of EV. If their experiments cannot be conclusive they should add this limitation in their Discussion Section.

Thank you for your comment. We understand your concerns that we do not directly show EV transfer of cel-miR-39, but a body of evidence to show EV release and EV containing miRNA. Nevertheless, we have now made a comment that we cannot be conclusive of EV transfer of cel-miR-39 in discussion section (pg17, second paragraph). 

- On the other side, focus has been put on one specific component of EV, miRNAs. However, to the best of my knowledge, EV also carry other component such as nucleic acids, proteins, and lipids, all of which might induce various cellular response in the recipient cells. Authors should discuss this point. In the same line, might EV carry the treatments that GenC received (glucose, PAN, LPS)? In this case the effects seen on recipient cells could also be due to treatment transfer, which is also interesting if possible. Please discuss this point with appropriate references.

Thank you for your comments. We understand and agree that other components of EVs may also induce cellular responses in recipient cells or potential treatment transfer. We have now added comments to this in discussion with appropriate references. (pg17, last paragraph). 

- Is the chosen pore size of the transwell system appropriate to reflect glomerular filtration barrier? Please justify this in the Methods Section.

Poor size was chosen to prevent direct cell migration and direct endothelial-podocyte contact. (pg6, 2nd paragraph). 

- Results regarding the role of miRNA-200c-3p are not sufficient in my opinion in order to establish a causal relationship between EV transfer of miRNA-200c-3p and VEGF production. First, the amount of miRNA-200c-3p transferred by EV on the one side, and by transfection on the other side, is not specified. The amounts are likely to be very different. Second, other components of EV might induce the decrease in podocyte VEGF production. Finally, podocyte impairment by either mechanism might result in VEGF decrease, rather than being the consequence of VEGF decrease. For these reasons, I suggest to authors to discuss this point, and to remove “via extracellular transfer of miRNA-200c-3p” from the title of the article, which is too assertive compared to the actual results of the experiments. The title might be “glomerular endothelial derived vesicles mediate podocyte dysfunction: a potential role for miRNAs” for example.

We understand your concerns and have changed title to “Glomerular endothelial derived vesicles mediate podocyte dysfunction: a potential role for miRNA”

---

## [Decision Letter · Decision Letter 1]

18 Feb 2020

PONE-D-19-29233R1

Glomerular endothelial derived vesicles mediate podocyte dysfunction: a potential role for miRNA

PLOS ONE

Dear Dr. Woollard,

Thank you for submitting your manuscript to PLOS ONE. After careful consideration, we feel that it has merit but does not fully meet PLOS ONE’s publication criteria as it currently stands. Therefore, we invite you to submit a revised version of the manuscript that addresses the points raised during the review  process :

There is a discrepancy between the quantification of EV secretion shown in the text of the "Results" section (3x10^10^ EV per 1x10^6^)  and the value given in the graph of fig 1 and in the "Discussion" section (3x10^9^ EV per 2x10^6 ^) . Could you compare your data to those in the literature?

We would appreciate receiving your revised manuscript by Apr 03 2020 11:59PM. To enhance the reproducibility of your results, we recommend that if applicable you deposit your laboratory protocols in protocols.io, where a protocol can be assigned its own identifier (DOI) such that it can be cited independently in the future. For instructions see: http://journals.plos.org/plosone/s/submission-guidelines#loc-laboratory-protocols

We look forward to receiving your revised manuscript.

Kind regards,

Jean-Claude Dussaule

Academic Editor

PLOS ONE

Reviewers' comments:

Reviewer's Responses to Questions

**Comments to the Author**

1. If the authors have adequately addressed your comments raised in a previous round of review and you feel that this manuscript is now acceptable for publication, you may indicate that here to bypass the “Comments to the Author” section, enter your conflict of interest statement in the “Confidential to Editor” section, and submit your "Accept" recommendation.

Reviewer #1: All comments have been addressed

Reviewer #2: All comments have been addressed

2. Is the manuscript technically sound, and do the data support the conclusions?

Reviewer #1: Yes

Reviewer #2: Yes

3. Has the statistical analysis been performed appropriately and rigorously? 

Reviewer #1: Yes

Reviewer #2: Yes

4. Have the authors made all data underlying the findings in their manuscript fully available?

Reviewer #1: Yes

Reviewer #2: Yes

5. Is the manuscript presented in an intelligible fashion and written in standard English?

Reviewer #1: Yes

Reviewer #2: Yes

6. Review Comments to the Author

Reviewer #1: The authors have adressed and justified all the minor comments that were pointed out at the time of their first submission to PLOS.

The manuscript is now suitable for publication.

Reviewer #2: The authors have addressed all my minor comments, and limitations and future insights in their findings are well pointed out in the Discussion Section.

7. PLOS authors have the option to publish the peer review history of their article (what does this mean?). If published, this will include your full peer review and any attached files.

Reviewer #1: Yes: Amelie Calmont

Reviewer #2: Yes: Nahid Tabibzadeh

---

## [Author Response · Author response to Decision Letter 1]

21 Feb 2020

There was a typo in results and discussion. Approximately 3x10(9) EV are secreted from 1x10(6) glomerular endothelial cells. This is broadly in-line with other published works (PMID 27010029)

---

## [Editor Report · Decision Letter 2]

27 Feb 2020

Glomerular endothelial derived vesicles mediate podocyte dysfunction: a potential role for miRNA

PONE-D-19-29233R2

Dear Dr. Woollard,

We are pleased to inform you that your manuscript has been judged scientifically suitable for publication and will be formally accepted for publication once it complies with all outstanding technical requirements.

With kind regards,

Jean-Claude Dussaule

Academic Editor

PLOS ONE
---

## [Editor Report · Acceptance letter]

10 Mar 2020

PONE-D-19-29233R2 

Glomerular endothelial derived vesicles mediate podocyte dysfunction: a potential role for miRNA 

Dear Dr. Woollard:

I am pleased to inform you that your manuscript has been deemed suitable for publication in PLOS ONE. Congratulations! Your manuscript is now with our production department. 

With kind regards,

on behalf of

Dr. Jean-Claude Dussaule 

Academic Editor

PLOS ONE